# Effect of Coating Pre-Treatment on Surface Recrystallization of DD6 Single Crystal

**DOI:** 10.3390/ma15197004

**Published:** 2022-10-09

**Authors:** Delin Liu, Jiaping Li, Xiaochao Jin, Rende Mu, Wenhui Yang

**Affiliations:** 1Aviation Key Laboratory of Science and Technology on Advanced Corrosion and Protection for Aviation Material, Beijing Institute of Aeronautical Materials, Beijing 100095, China; 2Joint Research Center for Extreme Environment and Protection Technology, School of Aerospace Engineering, Xi’an Jiaotong University, Xi’an 710049, China

**Keywords:** DD6 single crystal alloy, thermal barrier coatings, surface recrystallization, sandblasting pressure

## Abstract

Thermal barrier coatings (TBCs) are widely used to protect high-temperature components against harsh environments, such as extremely high temperatures. In this work, a second generation Ni-based single crystal superalloy (DD6) was treated in two ways: (1) via simple surface sandblasting under different pressures with no additional coating, and (2) through simple surface sandblasting under different pressures and then by applying NiCoCrAlYHf (HY5) coatings. The effects of pre-treatment (sandblasting) and the HY5 coating on the surface recrystallization of the alloy were thoroughly investigated. According to the results, both sandblasting pressure and the presence or absence of a coating significantly influence surface recrystallization. In particular, the critical sandblasting pressure for recrystallization increased the maximum recrystallization depth in both the coated and uncoated samples. Meanwhile, the recrystallization depth of the alloy with a coating was reduced compared to that without a coating. In addition, the number of recrystallized cells in the coated alloy was decreased, which indicated that the HY5 coating effectively reduced the degree of recrystallization.

## 1. Introduction

Thermal barrier coatings (TBCs) are widely used in aircraft engines to protect their superalloy turbine blades in high inlet-temperature environments. TBCs significantly prolong the components’ service life by greatly increasing the working temperature range and efficiency of high-temperature parts [1,2,3]. Sandblasting is one of the critical treatment processes before the turbine blades are covered with thermal barrier coatings, which can improve the bond strength between coating and substrate. However, sandblasting can result in residual stress accumulation and recrystallization of the blade surface during the operation, leading to plastic deformation and failure of the affected unit. Specifically, recrystallized boundaries of single crystal superalloys weaken the blades, subsequently deteriorating their high-temperature mechanical properties and service life [4,5,6]. Therefore, elucidating the influence of coating pre-treatment on the recrystallization of single crystal superalloys and their recrystallization behavior is an urgent task.

Several studies have shown that the thickening of the thermally grown oxide (TGO) layer is an important reason for the failure of TBCs. Nevertheless, some surface pre-treatment techniques of the bond coatings, such as sandblasting, polishing, and laser etching, can control the growth process of TGO [7,8,9,10]. However, these may cause the initiation and propagation of cracks that may exert a negative impact on the strength and microstructure of the bond coating [11,12]. Specifically, when developing and applying single crystal blades, solidification shrinkage caused by sand blowing, shaping, machining, and other processes can produce localized plastic deformation caused by internal stress [13]. Kromer et al. [14,15] evaluated the adherence between several controlled surface topographies obtained by grit-blasting and substrate. According to the results, the thermal spraying coating of a substrate with surface grit-blasting enhanced the bonding strength between the coating and the substrate. Xu et al. [10] studied the interfacial toughness and oxidation behavior of TBCs during thermal cycling and analyzed the effect of pre-treatment of the bond coatings by shot peening and sand blowing on the service life of TBCs. Xue et al. [16] investigated the effect of grit blasting, polishing, and electro-etched pre-treatment on the surface recrystallization of single crystal superalloys. Their results showed that grit blasting has the greatest influence on surface recrystallization, including the recrystallization depth and plastic deformation degree of the alloy.

Research on the crystallinity and thermal properties, as well as failure behaviors, of the TBCs, has also been carried out [17,18,19]. DD6 single crystal turbine blades were heat-treated in three ways and special attention was paid to the effects of coating on surface recrystallization of those blades. The results indicated that the coating changed the cellular recrystallization morphology of the original matrix, which could effectively reduce the degree of recrystallization [13]. Charpentier et al. [20] concluded that the initial polishing of ZrC/SiC specimens could increase the crystallinity of the oxide layer. This was because the initially rough surface was conducive to improving the hardness of the oxide layer, which might be due to a better adherence between the oxide layer and the substrate. Zhuo et al. [21] studied the recrystallization behavior of sandblasted single crystal superalloy cells via TEM and EBSD techniques. It was found that after annealing for 30 min at 1100 °C, the adjacent γ’ particles dissolved and merged into local irregular aggregates in the original matrix, and the formation of twins effectively reduced the energy in the recrystallization process. Dong et al. [22] explored the effects of water-blowing and heat exposure on the microstructure of DD6 alloy with TBCs. Specifically, the water-blown TBCs exhibited no recrystallization structure after the heat experiment. The combined method of adopting low laser energy and preheating at elevated temperatures can significantly lower the heat input into the single crystal (SX) matrix and, correspondingly, the interface stored energy, which can effectively lower the tendency for surface recrystallization after subsequent heat treatment [23]. At present, research on surface recrystallization of single crystal superalloys is mainly focused on the microstructure transformation of recrystallization zones [24,25], the factors leading to recrystallization itself [26,27] and the influence of recrystallization on the mechanical properties of the alloys [28,29]. However, few studies on how to reduce and eliminate recrystallization during the preparation of single crystal super-alloy blades have been reported, especially about the inhibition of the recrystallization of alloys by the addition of a coating. NiCoCrAlYHf (HY5) is a common metal protective coating material, and also a metal bonding layer material of thermal barrier coating, which can resist high-temperature oxidation corrosion and improve the bonding properties of the ceramic insulating layer and superalloy substrate [30,31]. Ni et al. [30] studied the isothermal oxidation behavior of NiCrAlY coatings, and the results showed that the oxidation resistance was the result of the aluminum contained in the NiCrAlY coatings, which could produce and maintain a continuous, dense, and adherent α-alumina layer upon high-temperature oxygen exposure. Shi et al. [31] found that the application of NiCrAlY and NiCoCrAlYHfSi coatings greatly improved the oxidation resistance of DD98M alloy. However, the effect of the coating on the surface recrystallization of DD6 single crystal blades is still not clear.

In this study, a fully heat-treated DD6 single crystal surface was treated by sandblasting, followed by thermal exposure. Special attention was paid to the effects of pre-treatment (sandblasting) and the presence or absence of an HY5 coating on the surface recrystallization of DD6 single crystal alloy. The recrystallization mechanism of the alloy was also explored.

## 2. Materials and Methods

### 2.1. Specimen Preparation

The material used in the experiment was the second-generation Ni-based single crystal superalloy DD6 with [001]-oriented rods, exhibiting excellent tensile and creep-rupture properties. The nominal chemical composition of the DD6 alloy before machining is given in Table 1. The heat treatment of the single crystalline bar consisted of a combined solution+aging procedure under the following temperature conditions: (1290 °C, 1 h) + (1300 °C, 2 h) + (1315 °C, 4 h, Air Cooling (AC)) + (1120 °C, 4 h, AC) + (870 °C, 32 h, AC). As shown in Figure 1, the microstructure of the alloy consisted of a matrix phase γ and a gridded strengthening phase γ’.

Rectangular specimens with dimensions 30 mm × 10 mm × 1.5 mm were machined from the DD6 single crystal alloy rod. The surfaces were polished with SiC sandpaper to remove metal debris and then ultrasonically cleaned in acetone and ethanol. Before deposition of the TBC coating, the specific surface of the specimen was grit-blasted under different sandblasting pressures. This process consisted of wet sandblasting with a grit size of about 125 μm at pressures of between 0 and 0.7 MPa, and the blasting time was 2 min. Specimens were subsequently cleaned in an ultrasonic bath for 10 min to reduce the amount of remnant grit particles on the surface and dried in compressed air. More details about the grit-blasting conditions can be found in Table 2. After grit-blasting, some of the specimens were coated with NiCoCrAlYHf (code HY5) via vacuum arc plating (see Table 1), to a thickness of about 20–40 μm, whereas the rest of the specimens remained uncoated.

### 2.2. Characterization Methods

The specimens were exposed to heat treatment in a high-temperature furnace at 850 °C (2 h) and 1100 °C (100 h). The recrystallization morphology of the specimens was observed by an FEI Novanano450 field-emission scanning electron microscope (SEM), (FEI Company, Hillsboro, OR, USA). In particular, the relationship between the recrystallization counts and the recrystallization depth versus the grit-blasting pressure was analyzed. The elemental distribution of the cross-sections of the alloys was examined via energy dispersive X-ray spectroscopy (EDS) (FEI Company, Hillsboro, OR, USA).

## 3. Results and Discussion 

### 3.1. The Effect of Sandblasting Pressure on Surface Recrystallization

Figure 2, Figure 3, Figure 4, Figure 5 and Figure 6 display the recrystallized morphologies of the samples with and without coating after treatment at different blasting pressures. According to Figure 2a,b, no recrystallization was observed within the specimen surface under a 0.1 MPa sandblasting pressure, whether the specimen had a coating or not. However, cellular recrystallization emerged in the uncoated specimen when the blasting pressure increased to 0.2 MPa. After sandblasting treatment at a certain pressure, the γ′ phase generated plastic deformation and the residual stress spread through the surface of the alloy. After high-temperature heat treatment, different cellular recrystallized structures were produced according to the surface residual stress induced by heating [27]. Although the heating temperature was lower than the solid-solution temperature of the γ’ phase (about 1250 °C), recrystallization occurred in the form of a cellular structure [32]. This cellular recrystallization is a form of recrystallization in which the interface between the crystallized zone and the matrix creates a large angle grain boundary, and harms the mechanical properties of single crystal materials [33,34].

As shown in Figure 3b, there was no cellular recrystallization in the coated specimen at a blasting pressure of 0.2 MPa, which indicated that the HY5 coating was exerting a specific inhibitory effect on the recrystallization of DD6. It is noteworthy that the recrystallization of superalloys generally nucleates on their surface rather than in the bulk. In this respect, the critical strain required for internal nucleation should be much larger than that of surface nucleation under the same temperature [35]. The main reason for recrystallization is that easy nucleation on the surface can reduce the newly formed interface, thereby decreasing the interface energy. Therefore, the coating has a specific inhibitory effect on recrystallization behavior, which will reduce the degree of recrystallization and depth.

Once the blasting pressure increased to 0.3 MPa, the maximum recrystallization depth of uncoated samples reached 12.27 μm, as shown in Figure 4a. Cellular recrystallization also occurred at the interface between the coating and the substrate (see Figure 4b), and the recrystallization depth was about 9.68 μm, which was significantly smaller than that in the uncoated specimen, thereby proving again the coating’s inhibitory effect on recrystallization. 

The recrystallization cell size and quantity of the uncoated specimen continued to increase with a further increase in the blasting pressure to 0.5 MPa, and the maximum recrystallization depth was 15.84 μm (Figure 5a), while the recrystallization depth of the coated specimen was about 13.28 μm (Figure 5b). However, the size and number of recrystallized cells in the uncoated specimen decreased as soon as the blasting pressure increased to 0.7 MPa, and the maximum recrystallization depth reached a value of about 12.36 μm (Figure 6a). Compared with the 0.5 MPa sandblasting pressure, the coated specimen’s cellular recrystallization size and number were also reduced. 

The results show that the internal structures of the samples after processing at different blasting pressures remained largely the same, consisting of network-like microstructures with a large number of cuboidal γ’ and γ phases. A comparison of Figure 2a,b revealed that there was an obvious γ-phase-free zone between the coating and the substrate of the coated sample, and the closer it was to the edge of the sample, the smaller the γ’ phase. Similar phenomena could also be found in other figures. As the blasting pressure increased, the thickness of the γ’-phase-free zone in the coated samples also increased. This was because the Al elements in the coating reacted with O to form an Al_2_O_3_ protective layer after exposure to high temperatures. At the same time, the Al atoms diffused out of the substrate, which led to a decrease in the γ’ phase through its transformation into a matrix phase γ.

Based on Figure 3, Figure 4, Figure 5 and Figure 6, the recrystallization cells consisted of many columnar γ′ precipitates in the γ matrix, and the columnar γ’ phase grew along the radial direction of a fan-shaped surface. From Figure 7, it can be seen that the surface area was a mixed oxide mainly composed of Al and O. In the recrystallization area, the O element content was higher than in other areas because the recrystallization began to form on the sandblasted surface and then spread to the center of the specimen, which brought a lot of O element into the substrate. In addition, the cellular recrystallized structure could be divided into three parts: the surface area where no γ′ phase existed, the central area where the γ′ inclusions were short, and the outer area near the grain boundaries with the longer γ′ species. These regions are referred to as area (1), area (2), and area (3) in Figure 7, respectively. This phenomenon was consistent with previously-reported results [32]. Recrystallization grains began to form on the sandblasted surface and grow from the surface to the center of the specimen. Previous studies have demonstrated that the presence of a solution of γ′ particles in front of the recrystallization grain boundary was essential for further grain growth. In the initial stage, because only a few γ′ precipitates were dissolved by the moving grain boundaries, low local solute concentration near the moving boundaries could not provide sufficient supersaturation for the nucleation of columnar γ′ precipitates. Thus, the columnar γ′ inclusions were absent in the surface area. In the course of the grain boundaries migration, a lot of γ′ precipitates were dissolved by the moving grain boundaries, which contributed to the nucleation and growth of the columnar γ′ precipitates. However, due to the high migration rate of grain boundaries, the diffusion of γ′-forming elements along the boundaries cannot provide sufficient elements for the continuous growth of the columnar γ′ precipitates. As a result, the columnar γ′ precipitates in the central area are short. Finally, the migration was slowed down when the recrystallization grain boundary entered the interior of the substrate, as the diffusion of γ′-forming elements along the migrating boundaries could continuously provide elements for the growth of long columnar γ′ precipitates. A similar phenomenon was observed in previous studies [32,36].

The maximum depth of the recrystallized region was afterward measured by drawing a vertical line, and the length was compared with the scale to get the thickness of the recrystallization layer. The statistical results on the recrystallization depth under different blasting pressures are shown in Figure 8. The number of recrystallized cells was counted along the length direction of the cross-sectional specimen (10 mm × 1.5 mm). With the increase in sandblasting pressure, recrystallization occurred, the depth of which increased gradually. Once the blasting pressure reached 0.5 MPa, the metal surface recrystallization depth and the number of recrystallized cells reached the maximum value. However, the recrystallization degree showed a downward trend as the blasting pressure increased. Compared with the uncoated specimens, the recrystallization depth and the number of recrystallized cells of the coated specimens were reduced, and the critical blasting pressure for recrystallization was 0.3 MPa, exceeding that for the uncoated specimens, as shown in Figure 9. The variation in the number of recrystallization cells with varying sandblasting pressure was consistent with the recrystallization depth. Based on Figure 8 and Figure 9, it could be seen that the HY5 coating had a significant inhibitory effect on the DD6 recrystallization.

The recrystallization of a metal surface is related to the deformation degree [37,38], inducing plastic deformation, and the critical deformation is the minimum level of deformation required for recrystallization to occur. At low sandblasting pressures, the degree of deformation on the metal surface was small and could not provide sufficient distortion energy for the recrystallization of the metal surface. With the increase in sandblasting pressure, the metal surface reached a certain degree of deformation, so the size of the recrystallized cells as well as their number and depth increased. 

### 3.2. Mechanisms of the Effect of the Coating on the Surface Recrystallization 

High-temperature oxidation is typical of uncoated single crystal superalloys when subjected to near-service temperatures. In this case, the Al and W elements will migrate to the surface of the alloy and combine with oxygen, leading to the dissolution of γ′ phases on the surface, which is conducive to recrystallization nucleation and grain boundary migration [39]. From the perspective of thermodynamics, the change in free energy for the formation of a new nuclear system is described as follows [13]:(1)$$\Delta G=n\Delta G_{v}+\eta n^{2/3}\sigma +n\xi v$$
where *n* is the number of atoms in the embryos, $\Delta G_{v}$ is the energy difference between the old and new phases for each atom, $\eta n^{2/3}$ is the nucleation surface area, *σ* is the average interfacial energy, and *ξv* is the strain energy of each atom. 

The formation of a new nucleus interface will increase the interface energy, and the energy required for a recrystallization core to form inside the material is much higher than that on the metal surface at the same temperature. The coating covers the blade substrate surface so that the recrystallization nucleation changes from external to internal, and the energy required for nucleation increases. Compared with the uncoated samples, the coated samples were difficult to recrystallize. In particular, the free energy of the DD6 superalloy was changed by coating, which made the blade surface less prone to recrystallization. In addition, monitoring of the recrystallization on the interface between NiCoCrAlYTa coating and nickel-based superalloy by [39] revealed that the sandblasting deformation layer covered by the coating did not recrystallize after vacuum diffusion treatment at 1000 °C for 6 h. However, when the treatment temperature increased to 1080 °C, obvious cellular recrystallization occurred in the sand-blown deformation layer covered by the coating. Moreover, Wang et al. [40] studied the recrystallization behavior in NiCoCrAlY-coated DS nickel-based superalloys during thermal aging, and their results indicated that the high-temperature holding time exerted a significant influence on the recrystallization. In this work, to prevent the occurrence of cellular surface recrystallization, the appropriate sandblasting pressure should be selected to reduce the degree of deformation of the blade surface; at the same time, the metal surface should be sprayed with a coating.

The single-crystal superalloy specimens covered with a NiCoCrAlY (code HY5) coating exhibited a dense oxide film on the coating surface after high-temperature treatment. According to the EDS scanning results (Figure 10), the superalloy surface layer had significant Al and O element content, indicating that this oxide film was mainly composed of alumina. The aluminum oxide film had a protective effect on the substrate and prevented it from being oxidized. At a blasting pressure of 0.2 MPa, the γ′ phase in the surface layer was dissolved by the high-temperature oxidation of the uncoated specimen, which promoted recrystallization. However, a protective aluminum oxide film formed on the surface of the coated samples at high temperatures. This was because of the oxidation resistance of HY5 coating, which protected the substrate surface from direct oxidization and, consequently, from recrystallization.

As the sandblasting pressure increased to 0.3 MPa, the deformation storage energy was sufficient to recrystallize the substrate surface of the coated specimen. However, compared with the uncoated specimen, the maximum recrystallization depth and the number of recrystallized cells of the coated specimen were relatively small regardless of the sandblasting pressure. This indicated that the HY5 coating had a specific inhibitory effect on the recrystallization of DD6 due to its ability to prevent oxidation of the substrate surface.

## 4. Conclusions

In this work, the effect of pre-treatment (sandblasting) on the recrystallization of DD6 single-crystal alloy coated with HY5 was investigated. Based on the findings, the following conclusions can be drawn:
(1)After sandblasting and thermal exposure for 100 h at 1100 °C, the critical sandblasting pressure for recrystallization of the coated substrate surface increased relative to that of uncoated specimens.(2)Compared with the uncoated specimens, the maximum recrystallization depth, and the number of recrystallized cells of the coated specimens were relatively small under the sandblasting pressure of 0.3–0.7 MPa.(3)The HY5 coating had a specific inhibitory effect on the recrystallization of DD6. On the one hand, the coated specimen nucleated on the substrate surface (interface), which required more interface energy; on the other, the coating could prevent the substrate surface from being directly oxidized, thereby avoiding the promotion of recrystallization during high-temperature oxidation.


## Figures and Tables

**Figure 1 materials-15-07004-f001:**
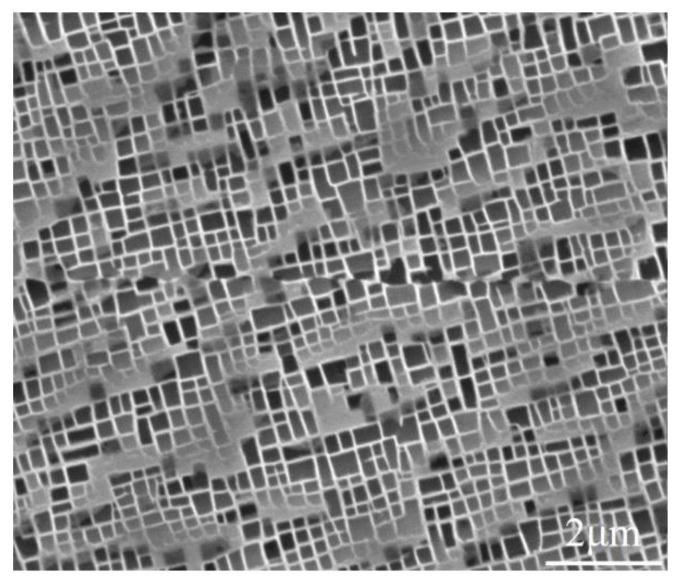
Microstructure of the DD6 single-crystal superalloy.

**Figure 2 materials-15-07004-f002:**
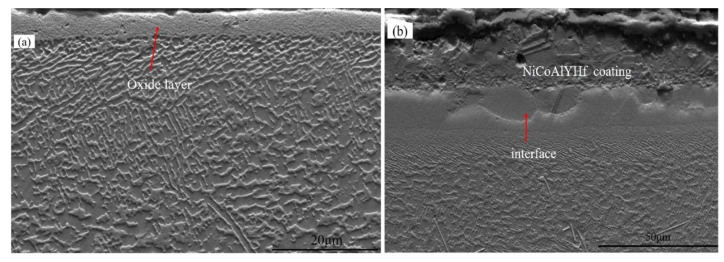
Specimen recrystallization morphology at 0.1 MPa sandblasting pressure in (**a**) uncoated and (**b**) coated alloys.

**Figure 3 materials-15-07004-f003:**
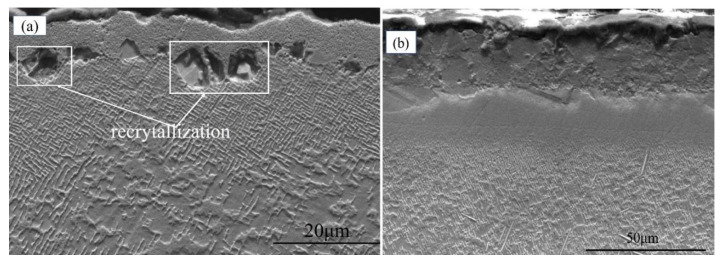
Specimen recrystallization morphology at 0.2 MPa sandblasting pressure in (**a**) uncoated and (**b**) coated alloys.

**Figure 4 materials-15-07004-f004:**
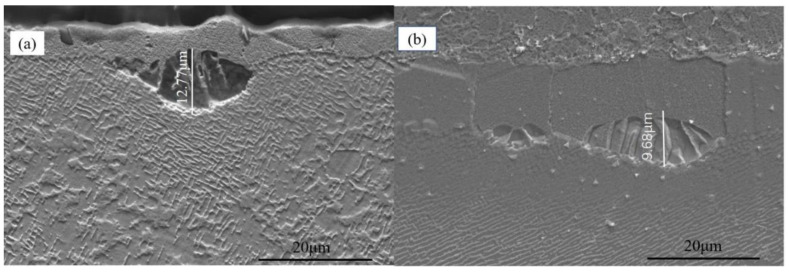
Specimen recrystallization morphology at 0.3 MPa sandblasting pressure in (**a**) uncoated and (**b**) coated alloys.

**Figure 5 materials-15-07004-f005:**
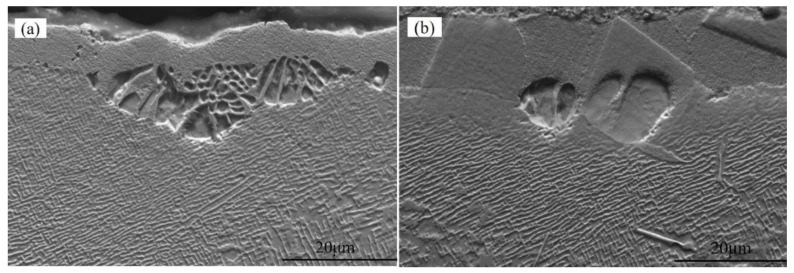
Specimen recrystallization morphology at 0.5 MPa sandblasting pressure in (**a**) uncoated and (**b**) coated alloys.

**Figure 6 materials-15-07004-f006:**
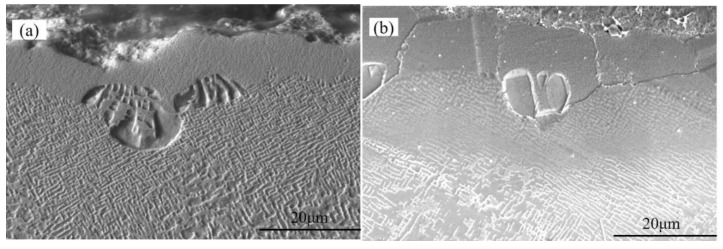
Specimen recrystallization morphology at 0.7 MPa sandblasting pressure in (**a**) uncoated and (**b**) coated alloys.

**Figure 7 materials-15-07004-f007:**
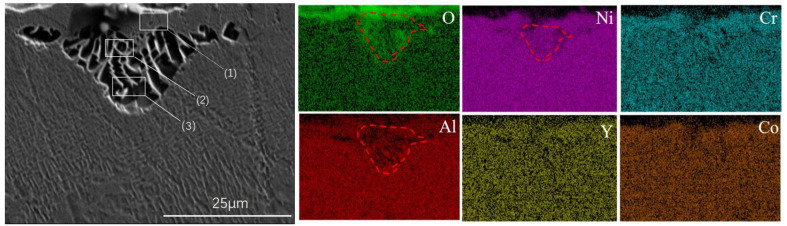
Microstructures and EDS results of a cellular recrystallized structure formed by exposure to high temperatures. (1) there is no columnar γ′ precipitates existing; (2) columnar γ′ precipitates is short; (3) columnar γ′ precipitates are long.

**Figure 8 materials-15-07004-f008:**
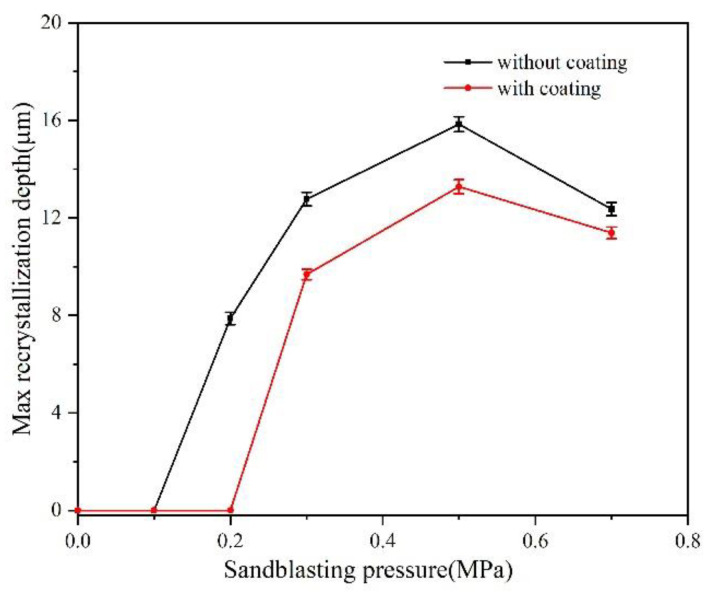
Relationship between the maximum recrystallization depth and sandblasting pressure.

**Figure 9 materials-15-07004-f009:**
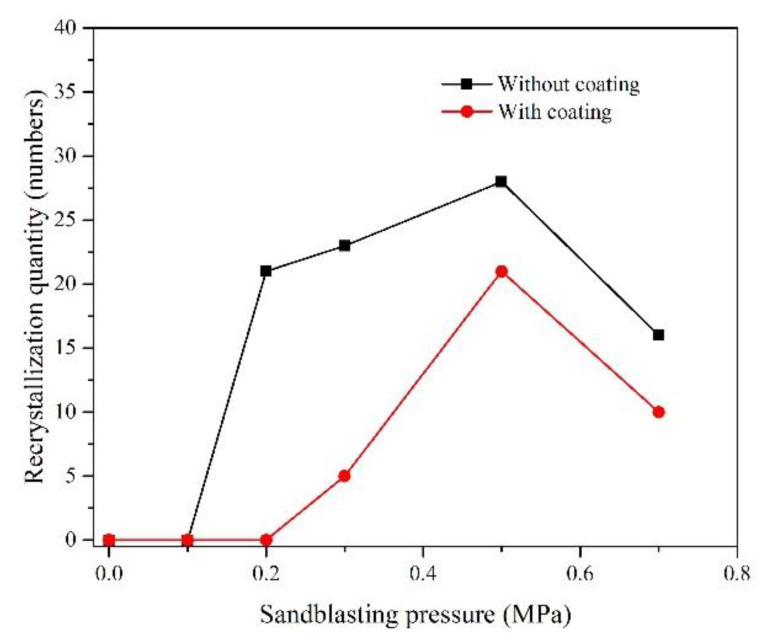
Relationship between the number of recrystallization cells and sandblasting pressure.

**Figure 10 materials-15-07004-f010:**
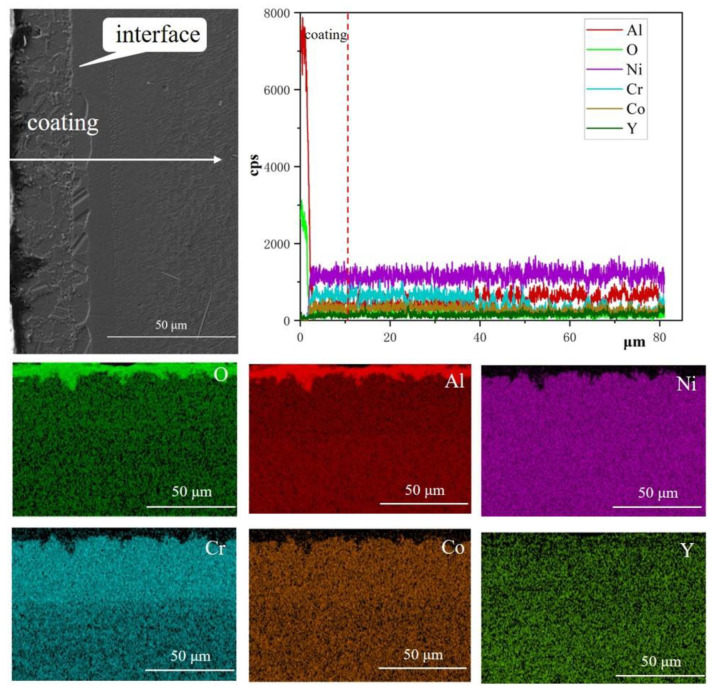
Elemental distribution of the alloys coated with HY5 from the surface to the inside after exposure to high temperatures.

**Table 1 materials-15-07004-t001:** Chemical compositions of the DD6 single crystal superalloy and HY5 coating (in wt.%).

Material	Co	Cr	Ta	Al	W	Hf	Re	Y	Ni
DD6	8.5~9.5	3.8~4.8	6.0~8.5	5.2~6.2	7.0~9.0	-	1.6~2.4	-	Bal.
HY5	10~15	18~23	-	8~12	-	0.2~0.6	-	0.1~0.5	Bal.

**Table 2 materials-15-07004-t002:** Sandblasting and coatings conditions of the single crystal superalloy specimens.

Sandblasting Pressure	Coating	Heat Treatment	Heat Exposure Experiment
0 MPa	Yes	850 °C, 2 h	1100 °C, 100 h
	No		
0.1 MPa	Yes		
	No		
0.2 MPa	Yes		
	No		
0.3 MPa	Yes		
	No		
0.5 MPa	Yes		
	No		
0.7 MPa	Yes		
	No		

## Data Availability

The data presented in this study are available on request from the corresponding author.

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
