# Peer review of "Effect of Coating Pre-Treatment on Surface Recrystallization of DD6 Single Crystal"

_materials, 2022, doi:10.3390/ma15197004_

Round 1

Reviewer 1 Report

This manuscript focuses on the effect of sandblasting and/or coating on the surface crystallisation of a single crystal Ni super alloy at 1100 ºC for 100 h. I think the topic is interesting and fits the scope of Materials. Nevertheless, I have some questions and comments:

1.     The authors associate the research on the crystallinity and mechanical properties of the TBCs to references 17 and 18, however, these refer to studies on hydroxyapatite coated titanium for implants. Could authors justify the selection of these references?

2.     There are several works on the recrystallisation behaviour of Ni single crystals thus, in my opinion, the innovative aspects of this work are not clear. I recommend authors to highlight the innovative characteristics of their study and include them in the Introduction.

3.     In section 2.1 it is referred ‘AC’ but not explanation is given for its meaning.

4.     In section 3.1, paragraph 3, authors mention ‘crystallisation’. Do you mean ‘recrystallisation’?

5.     In Figures 2-6 there seems to be a finer y’ distribution near the top region of the substrates. Why is that?

6.     Authors mention area (1) as the surface region where no y’ existed. Why is that? Does not this area correspond to the TGO?

7.     Authors mention a low diffusion rate of Al and Ti, however, according to Table 1 the substrate nor the coating have Ti in their composition.

8.     In the last sentence of section 3.1, authors say “the corresponding deformation is called the critical deformation above which the size and the number of recrystallised grains gradually decrease”. The critical deformation is the minimum level of deformation required for recrystallisation to occur, thus why would the number and the size of recrystallised grains decrease above this level?

9.     The authors state that the HY5 coating exerts a protective effect from oxidation and, consequently, from recrystallisation. How is this supported by data? In Figures 2-6b it is displayed a region depleted of y’ phase right beneath the HY5. Does not this represent dissolution of y’ phase? Could authors, please, explain these occurrences?

10.  As a further recommendation not, it is practical to add line number to the manuscript to ease the revision process.

Author Response

    Thanks very much for your constructive comments and suggestions, which are helpful to improve our manuscript. The main changes and modifications in the text have been highlighted with red fonts. We hope that the revised version has addressed all the comments.

Reviewer 2 Report

Delin Liu et al., explore the effect of Effect of Coating Pre-treatment Procession on Surface Recrystallization of DD6 Single Crystal. The manuscript looks very nice. It needs a minor revision on the basis of following comments.

1.      In the abstract section Kindly write complete name of DD6 and HY5.

2.      In introduction section, kindly explore the specific mechanical properties of the single crystals that can be improved by coating. For example, melting point, the ability to face the air resistance, corrosion effect of coating etc.

3.      How the deformation of DD6 single crystals is reduced by coating?

4.      The quality of figure 10 can be improved.

5.      DIO is missing in some references. For example, see reference number 10, 16, 21, 23, 24, 31, 36.

Author Response

(The authors gave the same response as above.)

Reviewer 3 Report

1. Please report the effect of the thickness of the HY5 coating. 

2. Only SEM can be misleading. There can be sample preparation artifacts and they may not give the full idea. Please report some surface characterization techniques, like SIMS, XPS, etc., with depth profiling. Please report the average value of several measurements 

Author Response

(The authors gave the same response as above.)

Round 2

Reviewer 1 Report

The revised manuscript is fine by me, overall. However, I recommend the authors to better address the issue regarding the presence of Ti. According to Table 1, the DD6 nor the coating have Ti in their chemical composition. Therefore, the explanation given by the authors regarding Ti diffusion does not make much sense (page 7, lines 9-16). Also, in page 2 line 42, the authors likely mean “NiCrAlY” (?) As a suggestion, in Figure 10, the composition profile graph should match the EDS compositional map colour for improved readability and interpretation.

Author Response

(The authors gave the same response as above.)

Reviewer 3 Report

please check the title 

Is procession word necessary

Author Response

Thanks very much for your suggestion, the “procession” has been removed from the title.